# Tackling Personalized Federated Learning with Label Concept Drift via Hierarchical Bayesian Modeling

**Xingchen Ma**          **Junyi Zhu**          **Matthew B. Blaschko**

Department of ESAT, PSI
KU Leuven, Belgium
`firstname.lastname@kuleuven.be`

## Abstract

Federated Learning (FL) is a distributed learning scheme to train a shared model across clients. One fundamental challenge in FL is that the sets of data across clients could be non-identically distributed. Personalized Federated Learning (PFL) attempts to solve this challenge. Most methods in the literature of PFL focus on the data heterogeneity that clients differ in their label distributions. In this work, we focus on label concept drift which is a broad but relatively unexplored area. We present a general framework for PFL based on hierarchical Bayesian inference and propose a variational inference algorithm based on this framework. We demonstrate our methods through empirical studies on CIFAR100 and SUN397. Experimental results show our approach significantly outperforms the baselines when tackling the label concept drift across clients.

## 1 Introduction

Since the introduction of FL in [12] in 2016, there has been increasing attention for this distributed learning setting, as it turns out to be a general framework to address the privacy concerns arise from many different areas. Despite its strength, there are also many challenges in the application of FL. One of them is the statistical heterogeneity of client data sets. Since clients sit in various environments, their data exhibit certain concepts that correlate with the local environments and deviate from each other. As summarized in [8], there exist different kinds of data heterogeneity. In this work, we focus on the so-called *label concept drift*. That is, the same label can have different features for different clients. As a result, the aggregated global model is not optimal for every individual client.

Personalized federated learning (PFL) methods target data heterogeneity issue and intend to achieve a better local utility via personalized models at the clients' side. Many of them view the learning process of PFL through optimization [5; 6; 7; 13], while a few [1; 4; 2] try to understand PFL as a posterior Bayesian inference problem. However, in the current literature on PFL, the most methods [1; 3; 7; 15; 13; 5; 6] focus on *label distribution skew*, i.e. clients have data corresponding to different labels. In this work we also take a Bayesian perspective, but aim at label concept drift and manage to capture the common trend while allowing each client to specialize in individual concepts.

**Our contributions** In this work we provide a variational Bayes framework for PFL to tackle label concept drift. Extensive experimental results show our method yields consistently superior performance to main competing frameworks.

## 2 Method

In this section, firstly we state the problem studied in this work, then we describe the proposed variational Bayesian framework for PFL under the label concept drift.

Workshop on Federated Learning: Recent Advances and New Challenges, in Conjunction with NeurIPS 2022 (FL-NeurIPS'22). This workshop does not have official proceedings and this paper is non-archival.

## 2.1 Problem Formulation

Let the data distribution of client $j$ be $P_j(x, y)$ where $x, y$ denote data and label, we present the definition of *label concept drift* following [8].

**Definition 2.1** (Label Concept Drift [8]). Let $J$ be the number of clients, label concept drift indicates that the conditional generating distributions $\{P_j(x|y)\}_{j=1}^J$ are different across different clients, but marginal distributions $\{P_j(y)\}_{j=1}^J$ are the same.

In FL a single shared model is used to fit on all clients' data, instead of doing that in PFL we aim to solve the following minimization problem:

$$\min_{\boldsymbol{w}_{1:J} \in \mathbb{R}^d} f(\boldsymbol{w}_{1:J}; \mathcal{D}) := \frac{1}{J} \sum_{j=1}^J f_j(\boldsymbol{w}_j; \mathcal{D}_j), \tag{1}$$

where $J$ is the number of clients, $\mathcal{D}_j$ is the set of data available in client $j$, $\boldsymbol{w}_{1:J}$ is a shorthand for the set of parameters $\{\boldsymbol{w}_1, \cdots, \boldsymbol{w}_J\}$, $\boldsymbol{w}_j$ is the personalized parameter for the $j$-th client and $f_j(\boldsymbol{w}_j; \mathcal{D}_j)$ is the empirical average loss function of the $j$-th client:

$$f_j(\boldsymbol{w}; \mathcal{D}_j) = \frac{1}{n_j} \sum_{i=1}^{n_j} l(\boldsymbol{x}_i^{(j)}, y_i^{(j)}; \boldsymbol{w}), \tag{2}$$

where $(\boldsymbol{x}_i^{(j)}, y_i^{(j)}) \in \mathcal{D}_j$ is one data point of client $j$, $l(\cdot, \cdot; \boldsymbol{w})$ is the loss function using weight parameter $\boldsymbol{w}$ and $n_j := |\mathcal{D}_j|$ is the number of data points on the $j$-th client.

## 2.2 The Augmented Joint Distribution

To develop a Bayesian framework, we need to obtain a posterior distribution for parameters which we are interested in. One way to obtain the posterior distribution $p(\boldsymbol{w}_j|\mathcal{D})$ is by performing Bayesian inference on the $j$-th client locally, i.e. $p(\boldsymbol{w}_j|\mathcal{D}) := p(\boldsymbol{w}_j|\mathcal{D}_j)$. Given a vague prior $p(\boldsymbol{w}_j)$, one disadvantage of this approach is the variance of $p(\boldsymbol{w}_j|\mathcal{D}_j)$ could be high if the number of data points on client $j$ is small. In another way, since all clients are running similar tasks, $\mathcal{D}_{\{1, \cdots, J\} \setminus j}$ should be able to provide information to form the posterior of $\boldsymbol{w}_j$. Therefore we introduce a *global variable* $\boldsymbol{w}$ such that all $\boldsymbol{w}_{1:J}$ depend on $\boldsymbol{w}$ and $\boldsymbol{w}$ captures the correlations between different clients, namely the common trend of the task. This also implies the conditional independence between $\boldsymbol{w}_i$ and $\boldsymbol{w}_j$:

$$p(\boldsymbol{w}_i|\boldsymbol{w})p(\boldsymbol{w}_j|\boldsymbol{w}) = p(\boldsymbol{w}_i, \boldsymbol{w}_j|\boldsymbol{w}). \tag{3}$$

Using Bayes' theorem, the augmented posterior distribution of $\{\boldsymbol{w}, \boldsymbol{w}_{1:J}\}$ is proportional to the product of the prior and the likelihood function, thus we have:

$$p(\boldsymbol{w}, \boldsymbol{w}_{1:J}|\mathcal{D}) \overset{3}{\propto} p(\boldsymbol{w}) \prod_{j=1}^J p(\boldsymbol{w}_j|\boldsymbol{w}) \exp\left(-f_j(\boldsymbol{w}_j; \mathcal{D}_j)\right), \tag{4}$$

where $p(\boldsymbol{w})$ is the prior of the introduced global variable, $f_j(\boldsymbol{w}_j|\mathcal{D}_j)$ is defined in Equation (2) and is proportional to the negative of the data log-likelihood on client $j$. Based on the above augmented model, two inference algorithms could be used. Maximum a Posteriori Probability (MAP) seeks a maximizer to the unnormalized posterior and avoids the intractable integration of the model evidence. However it can be difficult to select a set of hyper-parameters for the conditional priors. To mitigate this limitation, we present an algorithm based on the principle of maximizing the marginal likelihood in Section 2.3. The optimization will be conducted using variational expectation maximization.

## 2.3 Maximize the marginal likelihood

We assume an isotropic Gaussian as the form of conditional prior $p(\boldsymbol{w}_j|\boldsymbol{w})$ and use factorized variational approximation $q(\boldsymbol{w}_{1:J}) := \prod_{j=1}^J q_j(\boldsymbol{w}_j)$ to the true posterior distribution $p(\boldsymbol{w}_{1:J}|\mathcal{D})$. Additionally, the axis-aligned multivariate Gaussian is used as the variational family, that is, $q_j(\boldsymbol{w}_j) = \mathcal{N}(\boldsymbol{w}_j|\boldsymbol{\mu}_j, \boldsymbol{\Sigma}_j)$ and $\boldsymbol{\Sigma}$ is a diagonal matrix. To optimize these approximations $\{q_j(\boldsymbol{w}_j)\}_{j=1}^J$, we maximize the evidence lower bound (ELBO) of the marginal likelihood:

$$\text{ELBO}\left(q(\boldsymbol{w}_{1:J}), \rho_{1:J}^2, \boldsymbol{w}\right) = \sum_{j=1}^J \mathbb{E}_{q(\boldsymbol{w}_j)}[\log p(\mathcal{D}_j|\boldsymbol{w}_j)] - \text{KL}[q(\boldsymbol{w}_j) \| p(\boldsymbol{w}_j|\boldsymbol{w}, \rho_j^2)]. \tag{5}$$

The above ELBO can be optimized using variational expectation maximization through blockwise coordinate descent. To obtain the variational approximation $q(\boldsymbol{w}_j)$, we only need to maximize $\mathbb{E}_{q(\boldsymbol{w}_j)}[\log p(\mathcal{D}_j|\boldsymbol{w}_j)] - \mathrm{KL}[q(\boldsymbol{w}_j) \parallel p(\boldsymbol{w}_j|\boldsymbol{w}, \rho_j^2)]$ using the local set of data on client $j$.

After these local approximations have been formed, clients upload their own variational parameters to the server. Given these parameters, the server tries to optimize the ELBO in Equation (5) by updating the global variable $\boldsymbol{w}$. Simplifying Equation (5), we obtain the objective function on the server:

$$\mathrm{ELBO}(\rho_{1:J}^2, \boldsymbol{w}) \propto \sum_{j=1}^{J} \mathbb{E}_{q(\boldsymbol{w}_j)}[\log p(\boldsymbol{w}_j|\boldsymbol{w}, \rho_j^2)]. \tag{6}$$

Setting the gradient of Equation (6) w.r.t. $\boldsymbol{w}$ and $\rho_{1:J}^2$ to be zero, we obtain closed form for the optimization of the global variable:

$$\boldsymbol{w}^* = \frac{\sum_{j=1}^{J} \tau_j \boldsymbol{\mu}_j}{\sum_{j=1}^{J} \tau_j}, \tau_j := 1/\rho_j^2; \qquad D \cdot \rho_j^2 = \mathrm{Tr}(\boldsymbol{\Sigma}_j) + \parallel \boldsymbol{\mu}_j - \boldsymbol{w} \parallel^2, \tag{7}$$

where $D$ is the dimension of $\boldsymbol{w}$ and $\mathrm{Tr}(\boldsymbol{\Sigma}_j)$ is the trace of the variational variance-covariance parameter. We summarize the update rules in the following Algorithm 1.

---

**Algorithm 1** Variational Expectation Maximization for PFL (pFedVEM)

> **Input:** $T$ rounds, $\boldsymbol{\mu}_{1:J}^0$, $\boldsymbol{\Sigma}_{1:J}^0$, $\boldsymbol{w}^0$, $\rho_{1:J}^0$, sampling ratio $r$
> **Output** $\boldsymbol{w}_{1:J}^T$

1: **for** $t = 0$ to $T - 1$ **do**
2:      Server sends $\boldsymbol{w}^t$ to all clients
3:      ▷ *clients update their personalized variational distributions*                           ◁
4:      **for** $j = 1, \ldots, J$ **do**
5:          Update $\rho_j^{t+1}$ by $\rho_j^2 = (\mathrm{Tr}(\boldsymbol{\Sigma}_j^{t+1}) + \parallel \boldsymbol{\mu}_j^{t+1} - \boldsymbol{w}^t \parallel^2)/D$
6:          $(\boldsymbol{\mu}_j^{t+1}, \boldsymbol{\Sigma}_j^{t+1}) \in \arg\min_{(\boldsymbol{\mu}_j, \boldsymbol{\Sigma}_j)} \mathbb{E}_{q(\boldsymbol{w}_j)}[\log p(\mathcal{D}_j|\boldsymbol{w}_j)] - \mathrm{KL}[q(\boldsymbol{w}_j) \parallel p(\boldsymbol{w}_j|\boldsymbol{w}, \rho_j^2)]$
7:      Server select a random subset of clients $\mathcal{I}_t$ of size $\lfloor J \times r \rfloor$
8:      Each client $j \in \mathcal{I}_t$ sends its updated variational parameters $\boldsymbol{\mu}_j^{t+1}$ and $\rho_j^{t+1}$ to the server
9:      ▷ *server optimizes the global variable*                                     ◁
10:     $\boldsymbol{w}^{t+1} = \frac{\sum_{j\in\mathcal{I}_t} \tau_j^{t+1} \boldsymbol{\mu}_j^{t+1}}{\sum_{j\in\mathcal{I}_t} \tau_j^{t+1}}; \; \tau_j^{t+1} = 1/(\rho_j^{t+1})^2$

---

## 3 Experiments

To evaluate our methods, we target image classification tasks and compare our methods against baselines under various settings. We present results of the following frameworks: 1) Local: all clients train locally; 2) FedAvg [11]: all clients rely on an aggregated global model; 3) pFedPer [3]: the network consists of representation model and linear classifier, the representation model participates in collaboration while every client trains its own linear classifier; 4) pFedHN [14]: the server uses a hypernetwork and client-specific latent variable to generate a personal network for each client. 5) pFedME [16]: bi-level optimization of local models and the global model using Moreau envelops.

When applying our method, we follow the treatment in pFedPer, which considers the entire network comprising a representation model and a linear classifier, we only personalize the linear classifier layer while let the representation model to be trained via FedAvg to generate a general representation. Unifying the representation model could have additional profit for the representation model as it has many parameters while individual client has possibly limited data points. We conduct experiments on two variants of our method: 1) pFedVEM: our approach of variational inference. Every client has an estimation of the conditional distribution $p(\boldsymbol{w}_j|\boldsymbol{w})$ over its personalized linear classifier and the global variable $\boldsymbol{w}$ of linear classifiers $\boldsymbol{w}_{1:J}$ are aggregated w.r.t. the estimated distributions $\{p(\boldsymbol{w}_j|\boldsymbol{w})\}_{j=1}^{J}$. 2) pFedMAP: We also test a maximum posterior method based on our hierarchical Bayesian framework under non-informative prior of $\boldsymbol{w}$ and normal distributed $p(\boldsymbol{w}_j|\boldsymbol{w})$ with predefined variance $\rho_j$. We set $\rho_{1:J}$ to be the same to avoid a prohibitive tuning workload. The resulting algorithm is similar to Ditto [10].

|  | **CIFAR100** | | | **SUN397** | | |
|---|---|---|---|---|---|---|
| *# Clients* | 50 | 100 | 200 | 50 | 100 | 200 |
| *# Samples/Clients* | 1000 | 500 | 250 | 102 | 102 | 102 |
| Local | $36.4 \pm 0.3$ | $29.2 \pm 0.4$ | $22.8 \pm 0.3$ | $62.7 \pm 0.5$ | $62.6 \pm 0.6$ | $61.1 \pm 0.7$ |
| FedAvg | $52.2 \pm 0.1$ | $49.4 \pm 0.8$ | $45.8 \pm 0.1$ | $72.8 \pm 0.1$ | $75.4 \pm 0.2$ | $76.4 \pm 0.1$ |
| pFedME | $54.9 \pm 0.4$ | $48.9 \pm 0.5$ | $42.2 \pm 0.4$ | $85.1 \pm 0.5$ | $87.1 \pm 0.1$ | $87.0 \pm 0.2$ |
| pFedPer | $50.9 \pm 0.5$ | $44.6 \pm 0.3$ | $38.9 \pm 0.4$ | $84.4 \pm 0.3$ | $83.9 \pm 0.2$ | $83.7 \pm 0.2$ |
| pFedHN | $47.6 \pm 0.3$ | $46.2 \pm 0.2$ | $45.7 \pm 0.3$ | $75.9 \pm 0.5$ | $74.5 \pm 0.3$ | $78.3 \pm 0.3$ |
| **Ours** | | | | | | |
| pFedMAP | $58.8 \pm 0.3$ | $53.1 \pm 0.3$ | $47.3 \pm 0.4$ | $84.0 \pm 1.1$ | $86.6 \pm 0.4$ | $88.0 \pm 0.1$ |
| pFedVEM | $\mathbf{62.6 \pm 0.3}$ | $\mathbf{57.4 \pm 0.5}$ | $\mathbf{51.5 \pm 0.5}$ | $\mathbf{87.7 \pm 0.1}$ | $\mathbf{88.6 \pm 0.2}$ | $\mathbf{88.5 \pm 0.4}$ |

Table 1: Test accuracy ($\% \pm$ SEM) over 50, 100, 200 clients on CIFAR100 and SUN397. # Samples/Clients indicates the *expected data size of a client*.

## 3.1 Training protocol

To better present label concept drift, i.e. varying $P_j(x|y)$, we use hierarchical datasets CIFAR100 [9] and SUN397 [17], which contain superclasses and subclasses. We set the classification task to be superclass prediction, while for every client the data of each superclass is sampled from a random subclass and hence label concept drift is induced. To get closer to reality, we allow the number of observations by each client to be different by random sampling without replacement. When running the experiments the number of communication rounds is set to be 100.[1] We adopt the communication protocol proposed in pFedMe [16], i.e. at the beginning of each round, the server broadcasts the aggregated model to all clients, while only a subset of clients send their parameters back to the server. Each client has a probability of 0.1 to be sampled, we evaluate all frameworks with the number of clients $C \in \{50, 100, 200\}$.

We use a CNN network with five convolutional layers followed by a fully connected layer. In case a framework consists of two separate components, the convolutional layers block is used as the representation model and the fully connected layer as the linear classifer. We run all experiments five times on a cluster within the same container environment. The the same group of five random seeds has been used for local data sampling, clients sampling and parameters initialization. We present the mean and standard error.

## 3.2 Results

We evaluate all the frameworks over various settings, the results are given in Table 1. We observe that: 1) Although the label concept drifts among clients, FedAvg still outperforms Local. When the expected *# Sample/Clients* is large, e.g. on CIFAR100, FedAvg even outperforms some pFed frameworks, implying that although label concept drift across clients using a single global model is still adequate when all participants have sufficient data; 2) Our pFedMAP, which is lightweight and easy to implement, is competitive compared with previous PFL methods; 3) Our pFedVEM always achieves the best accuracy and performs significantly better than the baselines over all settings.

## 4 Conclusions

In this paper, we addressed the problem of personalized federated learning when label concept distributions differ between clients. We developed an evaluation framework for PFL with concept drift, and proposed a variational Bayes framework for PFL. Experimental results show our method yields consistently superior performance to main competing frameworks.

---

[1] 1) When conducting the local framework, we train the model for 100 epochs. 2) pFedHN limits the server to connect a single client at each iteration, so we let it connect with sampled clients in series at each round. Note that in reality this leads to a significant higher time complexity than other frameworks.

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
