# OpenReview forum: "Tackling Personalized Federated Learning with Label Concept Drift via Hierarchical Bayesian Modeling"
_NeurIPS.cc/2022/Workshop/Federated_Learning — FL-NeurIPS 2022 Oral_

### Official Review · Reviewer_YkEo · 2022-10-16

This work proposed a general framework for PFL based on hierarchical Bayesian inference.

Novelty: The novelty of the proposed work is limited. The proposed hierarchical Bayesian inference framework for personalized FL has been thoroughly discussed in [1] which has been published in this year NeurIPS, but the paper did does not cite it. A fair comparison and discussion are preferred.

[1] Self-Aware Personalized Federated Learning

Quality: The quality of this work is fair. I think you only need to send $\mu$ and $\rho$ in step 8 of Algorithm 1. No learning curves are provided. I think the paper should only keep algorithm 3 and 4 as the experiments are conducted with representation model.

Clarity: The clarity of this work is good.

Significance: The significance of this work is limited. It is difficult to follow the data partition procedure. A much more concise data partition procedure for label concept drift is preferred. You may also consider using datasets from domain adaption like Office datasaet.

I don't understand why does the proposed method is restricted to label concept drift. How does it perform in the label or quantity skew Non-IID setting, compared with other PFL baselines?

---

### Official Review · Reviewer_wWug · 2022-10-19
**Overall a good paper that tackles an interesting problem**

## Strengths:
* The paper tackles a relatively unexplored problem in FL (same labels, different features).
* The proposed framework is general and covers other personalized FL approaches.
* The evaluation seems to be reasonably solid.
## Weaknesses:
* The assumption that personalized models are conditionally independent with each other may not be realistic. It would be good to explain what happens when this assumption does not hold, as well as running experiments with mixed data distributions (instead of clean subclass partitioning).
* It would be much stronger to motivate this problem with real-world datasets. If two set of non-contradictory features map to the same label, a reasonably parametrized model should be able to learn both mappings. It would be good to show if this is the case using centralized learning on these datasets and see if a larger model could perform well with FedAvg.

---

### Decision · Program_Chairs · 2022-10-20

Accept (Oral)